# Petrography and Geochemistry of the Leucocratic Rocks in the Ophiolites from the Pollino Massif (Southern Italy)

**Giovanna Rizzo** [1], **Roberto Buccione** [1,*], **Michele Paternoster** [1,2], **Salvatore Laurita** [3], **Luigi Bloise** [4], **Egidio Calabrese** [4], **Rosa Sinisi** [5] **and Giovanni Mongelli** [1,5]

1   Department of Sciences, University of Basilicata, Campus Macchia Romana, 85100 Potenza, Italy;
    giovanna.rizzo@unibas.it (G.R.); michele.paternoster@unibas.it (M.P.); giovanni.mongelli@unibas.it (G.M.)
2   Istituto Nazionale di Geofisica e Vulcanologia (INGV), Sezione di Palermo, 90146 Palermo, Italy
3   TerraLab srl, Viale del Basento 118, 85100 Potenza, Italy; salvatore.laurita@terralab.eu
4   Parco Nazionale del Pollino, 85048 Rotonda, Italy; luigi.bloise@parcopollino.gov.it (L.B.);
    egidio.calabrese@parcopollino.gov.it (E.C.)
5   Institute of Methodologies for Environmental Analysis (CNR-IMAA), National Research Council of Italy,
    85050 Tito Scalo, Italy; rosa.sinisi@imaa.cnr.it
*   Correspondence: roberto.buccione@unibas.it

**Abstract:** In the Tethyan realm, leucocratic rocks were recognized as dikes and layers outcropping in the ophiolitic rocks of the Western Alps, in Corsica, and in the Northern Apennines. Several authors have suggested that the origin of leucocratic rocks is associated with partial melting of cumulate gabbro. Major and trace elements composition and paragenesis provided information about the leucocratic rocks genetic processes. This research aims at disclosing, for the first time, the petrographical and geochemical features of Timpa delle Murge leucocratic rocks, Pollino Massif (southern Italy), in order to discuss their origin and geodynamic significance through a comparison with other Tethyan leucocratic rocks. These rocks are characterized by high amounts of silica with moderate alumina and iron-magnesium contents showing higher potassium contents than plagiogranites, due to plagioclase alteration to sericite. Plagioclase fractionation reflects negative Eu anomalies indicating its derivation from gabbroic crystal mushes. The chondrite normalized REEs patterns suggest the participation of partial melts derived from a metasomatized mantle in a subduction environment. The results reveal some similarities in composition with other Tethyan leucocratic rocks, especially those concerning Corsica and the Northern Alps. These new data provide further clues on the origin of these leucocratic rocks and the Tethyan area geodynamic evolution.

**Keywords:** leucocratic rocks; geochemistry; fractional crystallization; ophiolite; geodynamic evolution; Pollino Massif; Southern Italy

## 1. Introduction

The petrography and the geochemistry of leucocratic rocks outcropping in the Pollino Massif ophiolitic sequence were analyzed. The origin of leucocratic rocks has been much debated since these rocks can provide important constraints on the evolution of the Earth's continental lithosphere [1].

Several authors [2–7] have proposed that these rocks derive from fractional crystallization of MORB-type basaltic magmas at low pressures, such as the Oman and Troodos ophiolitic sequences [8]. Other authors [9–11] have instead suggested an origin associated with partial melting of cumulate gabbro [10,12,13] while further authors [3,14] suggested that these rocks formed by presence of immiscible liquids found together with mafic melts.

It is important to note that geochemical data of some major oxides ($SiO_2$, $TiO_2$, $Na_2O$) and especially trace elements (Sr, Nb, Y, and REEs) are useful in order to distinguish between these different processes which led to the formation and evolution of granitoid rocks [11,12].

The purpose of this paper is to characterize, for the first time, with a petrographic and geochemical study, the leucocratic rocks of the ophiolitic rocks of the North Calabria Unit of the Pollino Massif, southern Italy. We also compare these rocks with those of Western Alps [15], Northern Apennines [16,17], and other Tethyan regions such as Turkey [18], Cyprus [1], Serbia [19], and Oman [20,21], in order to provide new constraints on the origin of these rocks within the general framework of the geodynamic evolution of the Tethyan realm.

## 2. Geological Framework and Field Relations

In the Calabrian-Lucanian Apennine, a sector of the southern Apennines, and along the northeast of the Pollino Massif, ophiolitic sequences crop out (Figure 1) [22–29]. The Calabrian-Lucanian Apennine formed in the Late Cretaceous-Oligocene age [30,31] and includes parts of the Liguride Complex [26,32–34].

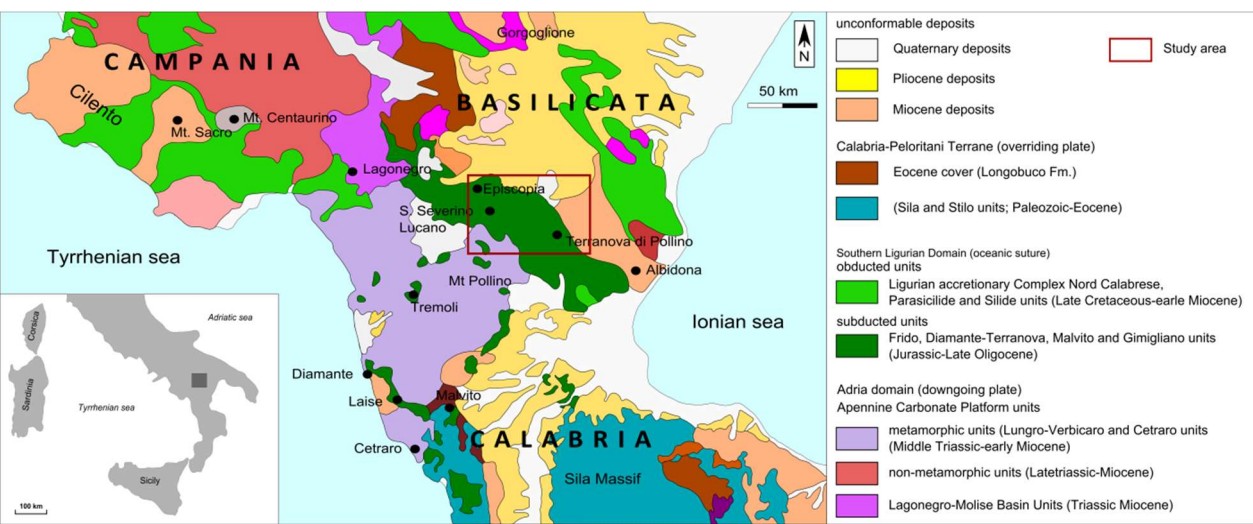

**Figure 1.** Geological map of Southern Apennines (modified after [26]).

The Liguride Complex includes, from bottom to top, the Northern Calabrian Unit, the Frido Unit, and the Crystalline Metamorphic Units [35,36]. The Northern Calabrian Unit consists of a pelitic matrix with blocks of ophiolitic rocks, pelagic sediments, and turbiditic successions [37]. The pelitic matrix is not foliated because the Northern Calabrian Unit is not affected by any orogenic metamorphism [28]. This unit shows sea-floor alteration [28] and any evidence of orogenic metamorphism. The Frido Unit consists of low-grade phyllites, calc-schists, and meta limestones with associated ophiolitic rocks [38]. Slivers of continental crust [24,25,31,36,37,39,40] occur in this unit. The association of continental crust slivers with ophiolites is commonly reported in ophiolite-bearing units of the Alpine-Apennine orogenic system [36]. The slivers of continental crust together with ophiolites have been involved in the subduction process responsible for the formation of the Liguride accretionary wedge [36]. Ophiolites of the Northern Calabrian Unit outcrop in the Timpa delle Murge and Timpa di Pietrasasso area [41]. In the Timpa delle Murge area ophiolites are composed of serpentinites, gabbro cumulates [39,42–44], and are associated with leucocratic rocks, forming a volumetrically minor part of an ophiolitic sequence (Figure 2). They are stratigraphically overlain by pillow lavas, hyaloclastites, and diabases. Serpentinites are characterized by a pseudomorphic texture and milonitic-cataclastic structures.

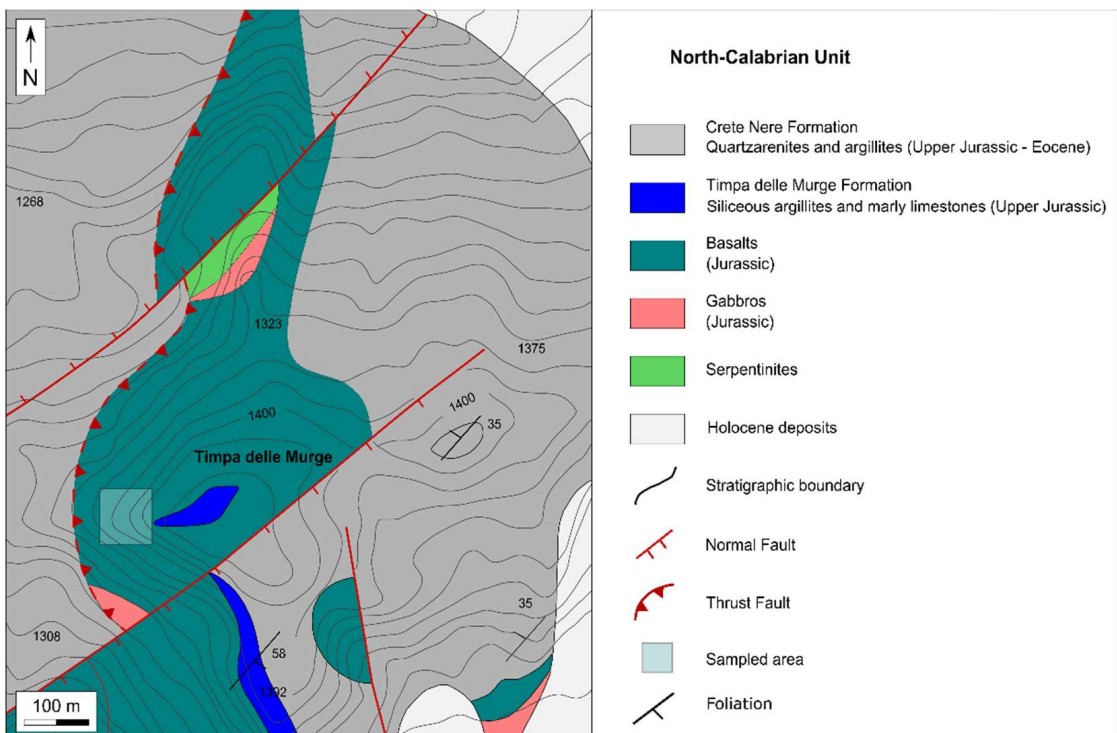

**Figure 2.** Geological schematic map of sampled area (Timpa delle Murge). Scale 1:10,000 (modified after [44]).

The intrusive complex of the ophiolitic sequences of the Northern Calabrian Unit is a minor part of the ophiolites from the Calabria-Lucania area and occurs as bodies not exceeding 1 km³ in volume [2].

Field observations indicate that the studied leucocratic rocks typically occur as flaps at the pseudo stratigraphic contact between gabbroic and basaltic bodies of the ophiolitic sequence of Timpa delle Murge. They appear in limited strips, and are medium to coarse grained, a few tens of meters thick and are light gray in color. The texture is massive and heterogeneous; quartz and plagioclase crystals can be identified in hand specimens (Figure 3).

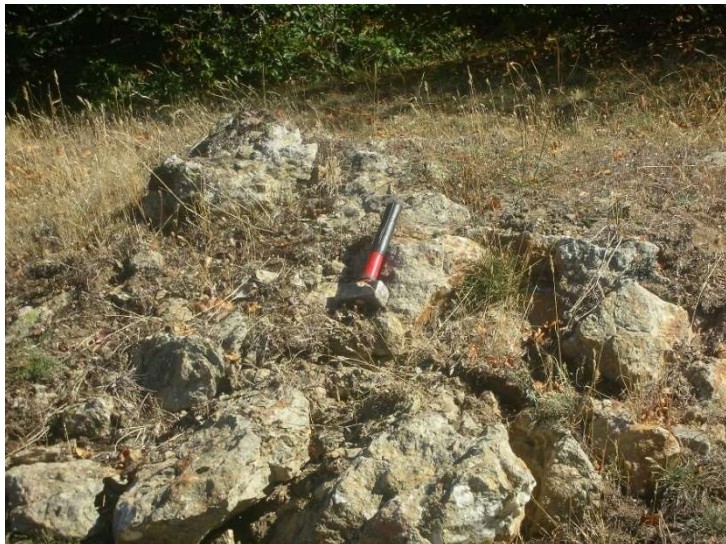

**Figure 3.** Occurrence of leucocratic rocks outcropping in the Timpa delle Murge area.

## 3. Materials and Methods

For this work, 16 samples of unweathered leucocratic rocks were collected at Timpa delle Murge. The specimens are leucocratic rocks of the Timpa delle Murge ophiolitic sequence, found in contact between gabbroic and basaltic bodies.

Petrographic and microstructural observations were carried out with a transmitted light microscope. Whole rock major ($SiO_2$, $Al_2O_3$, $Fe_2O_3$, MnO, MgO, CaO, $Na_2O$, $K_2O$, $TiO_2$, $P_2O_5$) and trace elements (Sc, Be, V, Ba, Sr, Y, Zr, Cr, Co, Zn, Ga, Ge, Rb, Nb, Ag, Sn, Cs, Hf, Ta, W, Tl, Pb, Th, U, REE) contents of the 16 samples were measured by using an inductively coupled plasma mass spectrometer (ICP-MS) at the Activation Laboratories (Ancaster, Canada) after sample powders digestion using a four acid attack. A 0.25 g sample was firstly digested using hydrofluoric acid, then using a nitric and perchloric acids mixture, before being heated in several ramping cycles using precise programmer-controlled heating that took the samples to incipient dryness. When the incipient dryness was obtained, the samples were brought back into solution using aqua regia and then they were analyzed with Varian ICP-OES and PerkinElmer ICP-MS instruments. Total loss on ignition (LOI) values were gravimetrically estimated after overnight heating at a temperature of 950 °C. Precision and accuracy are generally within 10%. Whole-rock mineralogy of samples was determined by X-ray powder diffraction (XRPD) at the Department of Sciences, University of Basilicata, Potenza, Italy, using a Siemens D5000 diffractometer with Cu-K$\alpha$ radiation and a 40 kV, 32 mA, and 0.02° (2θ) step size setup.

## 4. Results and Discussion

### 4.1. Petrography

The samples consist of leucocratic rocks and show equigranular, medium to coarse grained texture (Figure 4). The texture ranges from hypidiomorphic granular consertal (Figure 4a), where the boundaries between the quartz crystals are interdigitated and wedged together indicating simultaneous growth of the crystals, to granophyric, with vermicular intergrowths between quartz and plagioclase (Figure 4b). Coleman and Donato [45] suggest that the granophyric texture is a primary magmatic feature in granitic rocks. The mineral assemblage consists of quartz, plagioclase, and minor amounts of K-feldspar and secondary muscovite and generally these rocks show accessory minerals such as zircon, apatite, epidote, and opaque minerals [46,47]. Mafic minerals are primary biotite and secondary chlorite. Plagioclase crystals are euhedral and show deformation twins as well as an ondulose extinction and have albitic composition (Figure 4c). Quartz crystals show undulose extinction and incipient dynamic recrystallization along their boundaries, as shown by subgrains, new grains, and deformation lamellae (Figure 4d). K-feldspar is not interstitial since it is euhedral with tabular prismatic habit, often Carlsbad twinned with orthoclase. Muscovite forms tabular crystals associated with epidote and quartz and show undulose extinction and ductile deformations until two generations of muscovite are present. The first type of muscovite is igneous and forms large crystals with euhedral habit associated with epidote and quartz and shows undulose extinction and ductile deformations. The second type consists of small crystals associated with chlorite (Figure 4e). Biotite is present as brownish to greenish flakes, and contains zircon, apatite, and secondary epidote. Sericite and saussurite occur as secondary phases replacing plagioclase. Chlorite is present as pseudomorphs after biotite with inclusions of apatite, zircon, and opaques (Figure 4f). These rocks have undergone considerable deformation associated with metamorphic events, as clearly indicated by the microstructures observed in the major minerals as the simplectites that form as a breakdown product of K-feldspar during retrograde metamorphism.

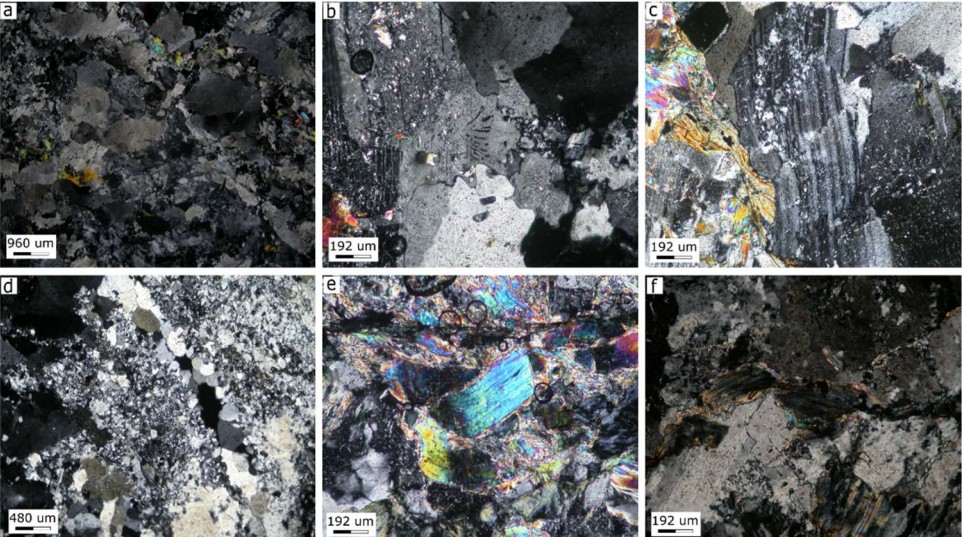

**Figure 4.** Petrographic major features of the analyzed leucocratic rocks: (**a**) hypidiomorphic granular consertal texture NX, 2×; (**b**) vermicular intergrowths between quartz and plagioclase NX, 10×; (**c**) deformed plagioclase as indicated by undulose extinction NX, 10×; (**d**) quartz with dynamic recrystallization as subgrains and newgrains NX, 4×; (**e**) muscovite with undulose extinction NX, 10×; (**f**) chlorite pseudomorphs after biotite and with inclusions of apatite, zircon and opaque minerals NX, 10×. NX = crossed polarizers.

Since mafic minerals do not exceed 10%, these rocks are classified, according to some authors [48–50] as trondhjemites, also evidenced in the Ab-An-Or triangular diagram (Figure 5).

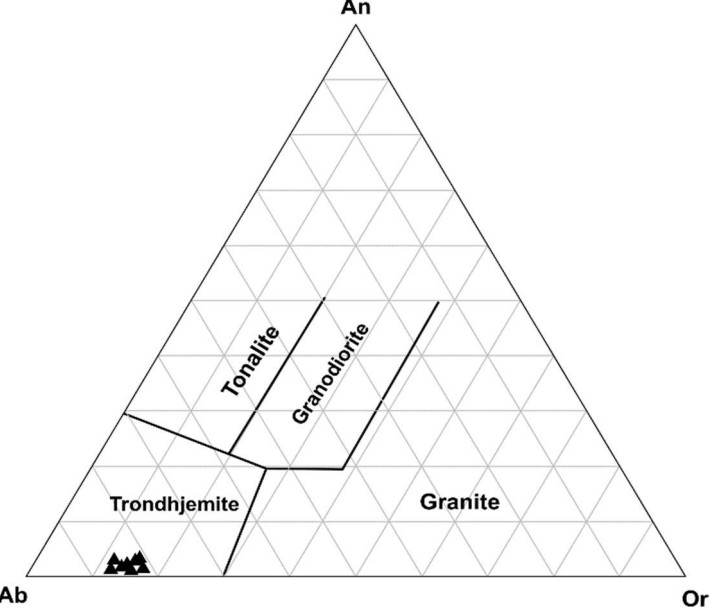

**Figure 5.** Normative Ab, An, Or diagram showing that the study samples plot in the Trondhjemite field.

The presence of chlorite, mica, and epidote crystals shows that these rocks are a result of metasomatism, probably related to fluid circulation near spreading centers [51].

### 4.2. Geochemistry

#### 4.2.1. Major Elements

The analyzed samples (Table 1) have a narrow range in $SiO_2$ between about 72.1 and 74.1 wt.% (median = 73.20 wt.%), $Al_2O_3$ varying between 13.20% and 13.89% (median = 13.66 wt.%) with lower abundances of $Fe_2O_3$ (median = 2.32 wt.%), $Na_2O$ (median = 4.90 wt.%), $K_2O$ (median = 1.45 wt.%), MgO (median = 0.90 wt.%), CaO (median = 0.42 wt.%), $TiO_2$ (median = 0.28 wt.%), $P_2O_5$ (median = 0.23 wt.%), and MnO (median = 0.03 wt.%).

**Table 1.** Whole-rock chemistry of the 16 samples.

| Element | d.l. | PL 1 | PL 2 | PL 3 | PL 4 | PL 5 | PL 6 | PL 7 | PL 8 | PL 9 | PL 10 | PL 11 | PL 12 | PL 13 | PL 14 | PL 15 | PL 16 |
|---|---|---|---|---|---|---|---|---|---|---|---|---|---|---|---|---|---|
| **%** | | | | | | | | | | | | | | | | | |
| $SiO_2$ | 0.01 | 74.00 | 72.98 | 73.00 | 73.25 | 72.28 | 72.13 | 74.01 | 73.15 | 74.05 | 72.24 | 73.15 | 73.66 | 74.07 | 74.08 | 73.80 | 73.10 |
| $Al_2O_3$ | 0.01 | 13.69 | 13.62 | 13.80 | 13.20 | 13.73 | 13.53 | 13.75 | 13.60 | 13.71 | 13.80 | 13.89 | 13.50 | 13.41 | 13.72 | 13.53 | 13.44 |
| MnO | 0.00 | 0.03 | 0.03 | 0.03 | 0.03 | 0.03 | 0.03 | 0.03 | 0.03 | 0.03 | 0.03 | 0.03 | 0.03 | 0.03 | 0.03 | 0.03 | 0.03 |
| MgO | 0.01 | 0.82 | 0.90 | 0.88 | 0.95 | 0.98 | 1.00 | 0.82 | 0.90 | 0.88 | 0.95 | 0.98 | 1.00 | 0.82 | 0.90 | 0.88 | 0.95 |
| CaO | 0.01 | 0.43 | 0.44 | 0.37 | 0.39 | 0.34 | 0.43 | 0.42 | 0.44 | 0.36 | 0.40 | 0,.34 | 0.42 | 0.43 | 0.44 | 0.38 | 0.39 |
| $Na_2O$ | 0.01 | 5.08 | 5.00 | 4.70 | 4.90 | 5.00 | 4.59 | 5.10 | 5.02 | 4.72 | 4.89 | 5.01 | 4.69 | 5.07 | 4.30 | 4.72 | 4.90 |
| $K_2O$ | 0.01 | 1.36 | 1.45 | 1.50 | 1.40 | 1.46 | 1.60 | 1.38 | 1.44 | 1.52 | 1.45 | 1.56 | 1.60 | 1.35 | 1.44 | 1.52 | 1.44 |
| $TiO_2$ | 0.00 | 0.32 | 0.30 | 0.28 | 0.28 | 0.24 | 0.30 | 0.31 | 0.30 | 0.27 | 0.28 | 0.24 | 0.29 | 0.35 | 0.31 | 0.27 | 0.27 |
| $P_2O_5$ | 0.01 | 0.24 | 0.20 | 0.30 | 0.20 | 0.19 | 0.20 | 0.23 | 0.21 | 0.32 | 0.21 | 0.19 | 0.25 | 0.24 | 0.22 | 0.32 | 0.23 |
| LOI | - | 1.72 | 1.75 | 1.74 | 1.90 | 1.89 | 2.10 | 1.20 | 1.74 | 1.84 | 1.92 | 1.88 | 1.12 | 1.77 | 1.85 | 1.76 | 1.91 |
| TOTAL | - | 99.97 | 99.35 | 99.54 | 99.60 | 98.34 | 98.40 | 99.78 | 99.17 | 99.99 | 98.55 | 99.48 | 100.09 | 99.99 | 99.93 | 99.46 | 99.04 |
| **ppm** | | | | | | | | | | | | | | | | | |
| V | 5 | 19 | 18 | 17 | 19 | 16 | 18 | 18 | 17 | 19 | 19 | 16 | 18 | 16 | 17 | 18 | 17 |
| Ba | 3 | 135 | 137 | 168 | 130 | 128 | 171 | 172 | 167 | 134 | 131 | 128 | 170 | 127 | 169 | 172 | 170 |
| Sr | 2 | 45 | 43 | 41 | 44 | 42 | 44 | 43 | 41 | 45 | 42 | 43 | 44 | 43 | 41 | 40 | 43 |
| Y | 2 | 8 | 10 | 11 | 8 | 12 | 10 | 9 | 11 | 10 | 12 | 8 | 12 | 10 | 11 | 8 | 10 |
| Zr | 4 | 147 | 145 | 130 | 147 | 121 | 144 | 148 | 146 | 133 | 127 | 147 | 123 | 145 | 132 | 145 | 142 |
| Cr | 20 | 190 | 230 | 260 | 200 | 260 | 280 | 191 | 220 | 250 | 270 | 254 | 232 | 195 | 281 | 264 | 190 |
| Co | 1 | 4 | 4 | 4 | 5 | 4 | 5 | 5 | 5 | 4 | 4 | 4 | 5 | 4 | 5 | 5 | 4 |
| Zn | 30 | 50 | 50 | 40 | 40 | 40 | 50 | 50 | 50 | 40 | 40 | 40 | 50 | 50 | 40 | 50 | 40 |
| Ga | 1 | 17 | 16 | 17 | 17 | 16 | 17 | 17 | 16 | 16 | 17 | 16 | 16 | 17 | 17 | 16 | 17 |
| Rb | 2 | 49 | 50 | 50 | 51 | 52 | 54 | 52 | 53 | 53 | 50 | 50 | 51 | 54 | 54 | 48 | 50 |
| Nb | 1 | 8 | 8 | 10 | 9 | 8 | 10 | 10 | 8 | 8 | 9 | 8 | 9 | 8 | 10 | 10 | 8 |
| Ta | 0.1 | 0.8 | 0.82 | 0.8 | 0.75 | 0.7 | 0.7 | 0.71 | 0.8 | 0.81 | 0.72 | 0.81 | 0.82 | 0.81 | 0.74 | 0.7 | 0.71 |
| Pb | 5 | 13 | 14 | 12 | 15 | 10 | 16 | 15 | 16 | 10 | 12 | 14 | 12 | 13 | 11 | 14 | 14 |
| La | 0.1 | 38.2 | 40 | 37 | 39 | 35.8 | 40.4 | 35.7 | 36.4 | 39 | 40 | 38 | 39.2 | 37.5 | 38.6 | 40.05 | 37 |
| Ce | 0.1 | 73.8 | 77 | 75 | 70 | 69.4 | 78 | 70 | 74 | 77 | 75 | 75 | 69.5 | 70 | 71 | 76 | 75 |
| Pr | 0.05 | 8.37 | 9.2 | 8.5 | 8 | 7.5 | 8.68 | 9.1 | 8.4 | 7.7 | 7.5 | 8.7 | 8.1 | 9.1 | 9.15 | 8.3 | 9.2 |
| Nd | 0.1 | 32.9 | 33 | 29 | 32 | 29.8 | 33.8 | 32.7 | 32.8 | 33.15 | 29.2 | 32 | 29.65 | 33.7 | 28.8 | 29.9 | 30.1 |
| Sm | 0.1 | 6.5 | 6 | 6.3 | 5.9 | 5.6 | 6.5 | 6.05 | 5.87 | 6 | 5.9 | 5.7 | 6.5 | 6.35 | 6.12 | 6.3 | 5.7 |
| Eu | 0.05 | 0.8 | 1 | 0.98 | 0.99 | 1.01 | 1.08 | 0.97 | 0.99 | 0.99 | 0.99 | 0.99 | 1.01 | 0.85 | 1.04 | 0.97 | 0.99 | 1.11 |
| Gd | 0.1 | 4.1 | 4.2 | 4 | 4.3 | 4 | 4.5 | 4 | 4.2 | 4.05 | 4.51 | 4.12 | 4.43 | 4.15 | 4.31 | 4.1 | 4.4 |
| Tb | 0.1 | 0.5 | 0.5 | 0.5 | 0.5 | 0.5 | 0.5 | 0.5 | 0.5 | 0.5 | 0.5 | 0.5 | 0.5 | 0.5 | 0.5 | 0.5 | 0.5 |
| Dy | 0.1 | 1.7 | 2 | 2.2 | 2.3 | 2.3 | 2.4 | 2.4 | 1.8 | 2.2 | 2.4 | 2.3 | 2.3 | 2.3 | 1.7 | 1.9 | 1.9 |
| Ho | 0.1 | 0.3 | 0.3 | 0.4 | 0.4 | 0.4 | 0.4 | 0.4 | 0.3 | 0.5 | 0.4 | 0.4 | 0.4 | 0.3 | 0.3 | 0.5 | 0.5 |
| Er | 0.1 | 0.6 | 0.7 | 0.6 | 0.8 | 0.9 | 0.8 | 0.7 | 0.8 | 0.7 | 0.6 | 0.6 | 0.9 | 0.9 | 0.8 | 0.8 | 0.9 |
| Tm | 0.05 | 0.07 | 0.08 | 0.09 | 0.07 | 0.11 | 0.1 | 0.08 | 0.09 | 0.09 | 0.08 | 0.11 | 0.12 | 0.12 | 0.09 | 0.08 | 0.07 |
| Yb | 0.1 | 0.4 | 0.3 | 0.6 | 0.3 | 0.6 | 0.6 | 0.5 | 0.4 | 0.6 | 0.3 | 0.5 | 0.5 | 0.4 | 0.3 | 0.6 | 0.6 |
| Lu | 0.04 | 0.06 | 0.06 | 0.09 | 0.08 | 0.09 | 0.09 | 0.06 | 0.06 | 0.08 | 0.08 | 0.08 | 0.09 | 0.09 | 0.08 | 0.08 | 0.06 |
| Ce/Ce* | - | 0.97 | 0.94 | 0.99 | 0.93 | 0.99 | 0.98 | 0.91 | 0.99 | 1.04 | 1.01 | 0.97 | 0.91 | 0.89 | 0.89 | 0.98 | 0.95 |
| Eu/Eu* | - | 0.47 | 0.61 | 0.60 | 0.60 | 0.65 | 0.61 | 0.60 | 0.61 | 0.61 | 0.59 | 0.64 | 0.48 | 0.62 | 0.58 | 0.60 | 0.68 |
| (La/Yb)ch | - | 64.53 | 90.10 | 41.67 | 87.85 | 40.32 | 45.50 | 48.25 | 61.49 | 43.92 | 90.10 | 51.36 | 52.98 | 63.35 | 86.95 | 45.11 | 41.67 |
| (Gd/Yb)ch | - | 8.31 | 11.35 | 5.40 | 11.62 | 5.40 | 6.08 | 6.48 | 8.51 | 5.47 | 12.18 | 6.68 | 7.18 | 8.41 | 11.64 | 5.54 | 5.94 |

d.l. = detection limits. Ce/Ce* = $[Ce_n/\sqrt{(La_n \cdot Pr_n)}]$; Eu/Eu* = $[Eu_n/\sqrt{(Sm_n \cdot Gd_n)}]$.

#### 4.2.2. Trace Elements

Regarding trace elements (Table 1), the most abundant are Ba (median = 152 ppm), Zr (median = 145 ppm), and Cr (241 ppm) while further trace elements such as V (median = 18 ppm), Sr (median = 43 ppm), Y (median = 10 ppm), Co (median = 4 ppm), Zn (median = 45 ppm), Ga (median = 17 ppm), Rb (median = 51 ppm), Nb (median = 8.5 ppm), Pb (median = 13.5 ppm), and Th (median = 12.21 ppm) show lower abundances. The analyzed samples show a $\sum$REE median of 166.83 ppm.

*4.3. Geochemical Features and Genetic Interpretation*

The Na₂O content in analyzed leucocratic rocks is consistent with those of mid-ocean ridge basalts (MORB) as defined by Silantyev et al. [52] ($\leq$8.0 wt.%). However, the relatively high potassium content indicates that the studied rocks experienced a secondary alteration that is testified by the muscovite/sericite occurrence [53].

The SiO₂ vs. Na₂O+K₂O diagram [48,54] (Figure 6a) indicates that the studied samples may be classified as sub-alkaline in composition characterized by Na₂O values from 4.59 to 5.08 wt.% which matches the high modal abundance of plagioclase with albitic composition observed in these rocks.

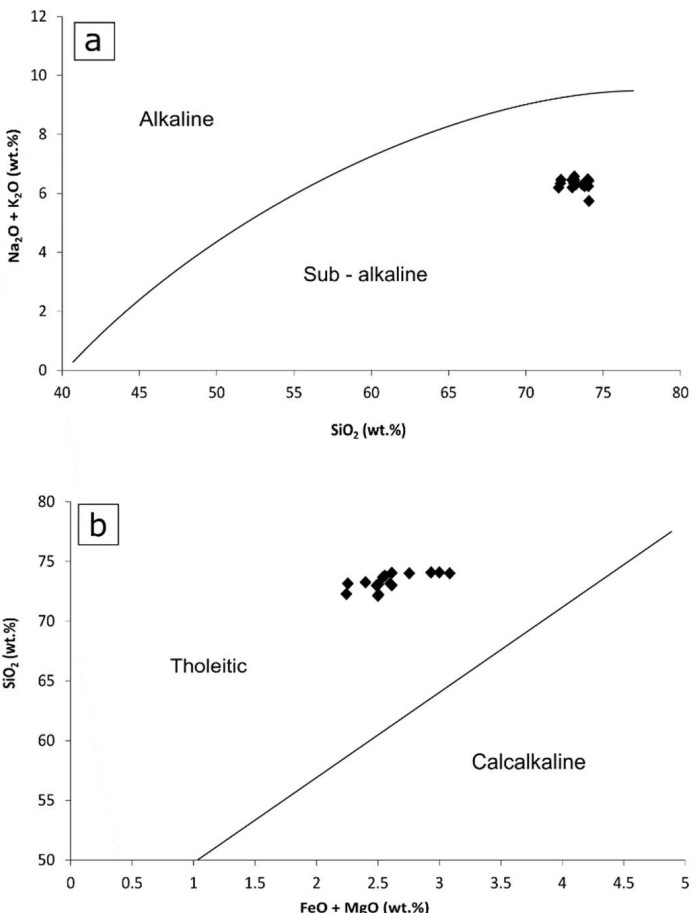

**Figure 6.** (**a**) SiO₂ (wt.%) vs. Na₂O+K₂O (wt.%) variation diagram (modified after [48,54]); (**b**) FeO/MgO (wt.%) vs. SiO₂ (wt.%) variation diagram (modified after [55]).

The FeO/MgO vs. SiO₂ diagram [55] (Figure 6b) shows that all samples fall in the tholeiitic compositional field with FeO/MgO values ranging from 2.26 to 3.10. Finally, in the SiO₂ vs. K₂O diagram [56] (Figure 7) the analyzed rocks, showing K₂O between 1.36 and 1.60 wt.%, plot into the continental trondhjemite field.

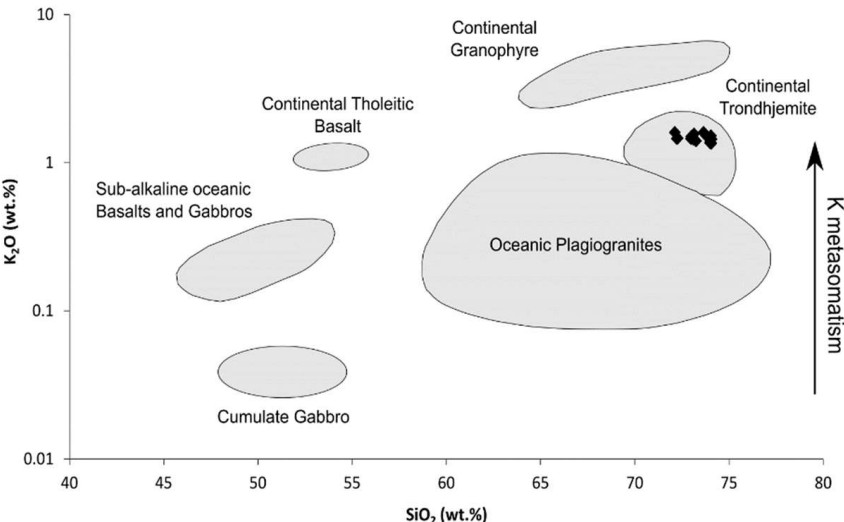

**Figure 7.** SiO$_2$ (wt.%) vs. K$_2$O (wt.%) variation diagram. All studied leucocratic rocks samples fall in the continental trondhjemite field. Modified after [56].

Trace element contents were normalized to MORB composition defined by Pearce et al. [57]. Figure 8 shows that the analyzed rocks have high content of K and Ba of about 10 times the MORB, Rb and Th of about 70–80 times those of MORB, enrichments in Ta, Nb, Ce, P, Zr, Hf, Sm, between 1 and 10 times the MORB, and depletions in Sr, Y, and Yb compared to the MORB. The depletion in Sr resulting from Figure 8 may indicate fluid leaching but may alternatively be interpreted as the result of trondhjemites fractionation retained in gabbroic mushes, and thus related to the negative Eu anomaly (Figure 9) [58]. The relative enrichment in K and Rb depends on the presence of K-feldspar whereas the moderate negative Ba anomaly may be due to the low degree of alteration of the samples [51,59].

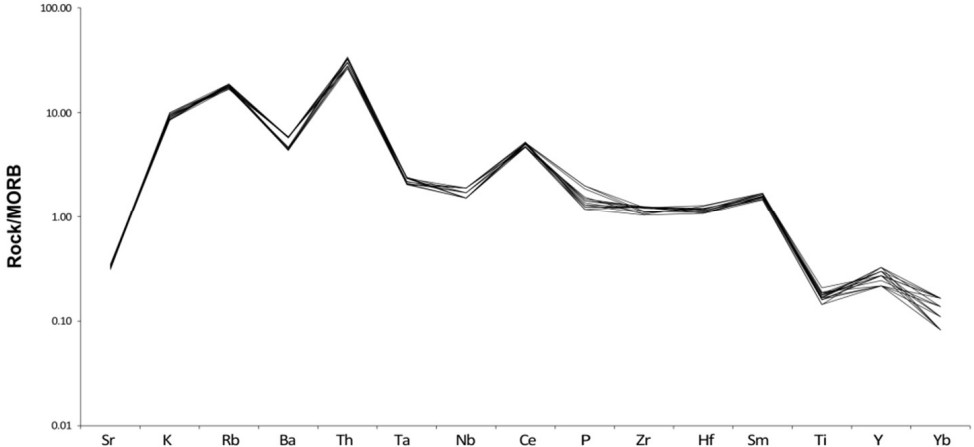

**Figure 8.** Trace elements MORB normalized spider diagram. Normalization was undertaken with respect to values by Pearce et al. [57].

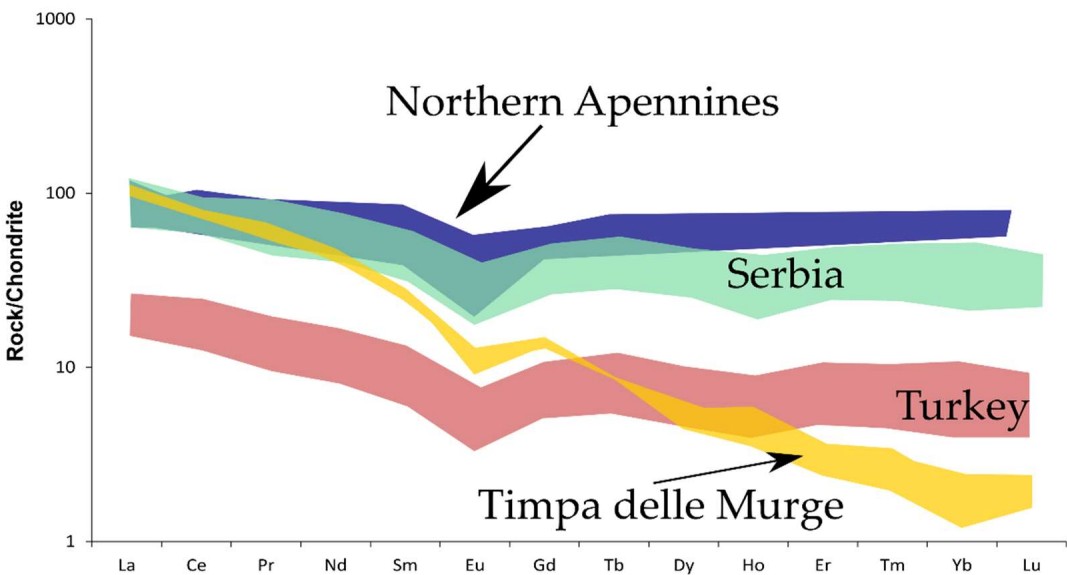

**Figure 9.** Chondrite normalized REEs patterns of leucocratic rocks of Timpa delle Murge and other Tethyan plagiogranites. Normalization was undertaken with respect to values by McDonough et al. [60].

Chondrite normalized REEs patterns (Figure 9, normalization values from [60]) lack anomalies in Ce for all samples (Ce/Ce* median = 0.97). In contrast, negative Eu anomalies were observed for all samples with a median of 0.61 and a range of 0.47–0.68, consistent with fractional crystallization of plagioclase from precursor highly crystalline crystal mushes [59]. The role of plagioclase fractionation is clearly evident in the increasing depletion of Eu during magma evolution from gabbroic to plagiogranite as occurs in many arc and rift-related granites rocks [17]. In Table 1, further fractionation indices such as $(La/Yb)_{ch}$ and $(Gd/Yb)_{ch}$ were calculated exhibiting median values 52.17 and 6.93, respectively. The resulting chondrite normalized REEs patterns are very similar for all studied samples showing, in particular, a major enrichment in light over heavy REEs. The LREEs concentrations (about 100 La) are higher than those of the HREEs by almost two orders of magnitude. The light and middle REEs (La-Tb) show less variation between samples than the heavy REE (Dy-Lu). The homogeneous chondrite normalized REEs patterns of the leucocratic rocks of Timpa delle Murge exhibit higher $(La/Yb)_{ch}$ values with depletion of MREEs and HREEs with respect to those of other Tethyan plagiogranites (Figure 9). This REEs pattern is typical of leucocratic and granitoid rocks such as tonalites and trondhjemites suggesting we are in the presence of differentiated rocks that originated by mantle partial melting in a subduction environment, in agreement with what was reported, for instance, by [10–12,14]. The $(Gd/Yb)_{ch}$ values indicate that other phases (such as amphibole and/or garnet) were fractionated pointing to an origin in a thickened continental crust.

A comparison among the samples of Timpa delle Murge with those of several Tethyan realm plagiogranites of ophiolitic complexes [1,11,12,17,19,21,42,52,61], was made. These rocks are characterized by $SiO_2$ contents showing a median between 65 and 70 wt.%, $Al_2O_3$ with a median of about 15 wt.%, $Fe_2O_3$ in the 0 to 5–6 wt.% range, $Na_2O$ ranging from 3 to 10 wt.%, $K_2O$ from 0 to 4 wt.% and MgO in the 0–7 wt.% range (Table 1). Furthermore, Figure 10 displays a box and whiskers plot for major oxides for both the analyzed samples from Timpa delle Murge as well as for other Tethyan plagiogranites.

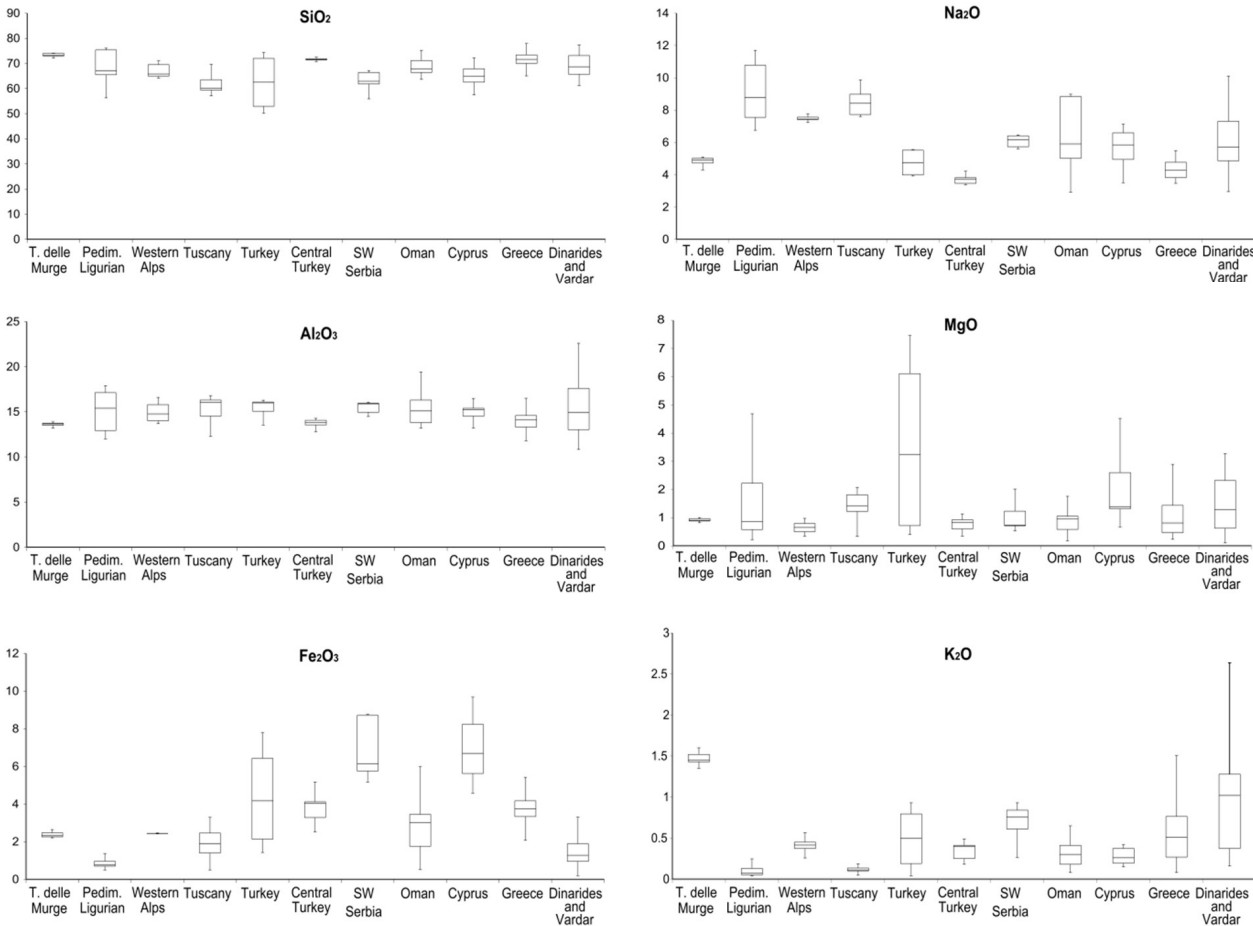

**Figure 10.** Box and whiskers plot for major oxides (SiO$_2$, Al$_2$O$_3$, Fe$_2$O$_3$, Na$_2$O, MgO, K$_2$O) in considered samples and further Tethyan plagiogranites rocks. The values are expressed in %.

The SiO$_2$ median value (73.20 wt.%) of the Timpa delle Murge leucocratic rocks is quite similar to those of the silica-rich Central Turkey (median = 71.53 wt.%) and Hellenic (median = 71.60 wt.%) plagiogranites. The Al$_2$O$_3$ median value (13.66 wt.%) of Timpa delle Murge leucocratic rocks is similar to that of both Central Turkey (13.83 wt.%) [18] and Dinarides plagiogranites (13.60 wt.%) [53] but lower than in other Tethyan plagiogranites such as the Vardar ones (14.22 wt%) [53]. Finally, it was observed that Timpa delle Murge leucocratic rock samples show slightly higher K$_2$O median values (1.54 wt.%) compared to the other localities reaching a maximum median of 1.02 wt.% in the Dinarides and Vardar plagiogranites. Additionally, the median values of Na$_2$O in the Timpa delle Murge samples (4.90 wt.%) are very similar to the median values of Turkish (4.75 wt.%) and Hellenic plagiogranites (4.27 wt.%), that show the lowest concentrations among the Tethyan plagiogranites (Figure 10) [18,51,53].

We extended our comparison to the ternary AFM diagram (Figure 11a) [15,17]. There is a considerable difference among the plagiogranites, in particular regarding the relative abundances of FeO$_{Tot}$ and Na$_2$O + K$_2$O. The plagiogranites with the higher values of FeO$_{Tot}$ are those of the Verné Association whereas lower values of the FeO$_{Tot}$ belong to the plagiogranites of Sestri Voltaggio zone, and some samples from Corsica and Northern Apennines associations. FeO$_{Tot}$ concentrations of Timpa delle Murge leucocratic rocks show intermediate values which overlap quite precisely with the plagiogranites of Corsica suggesting similar genetic conditions.

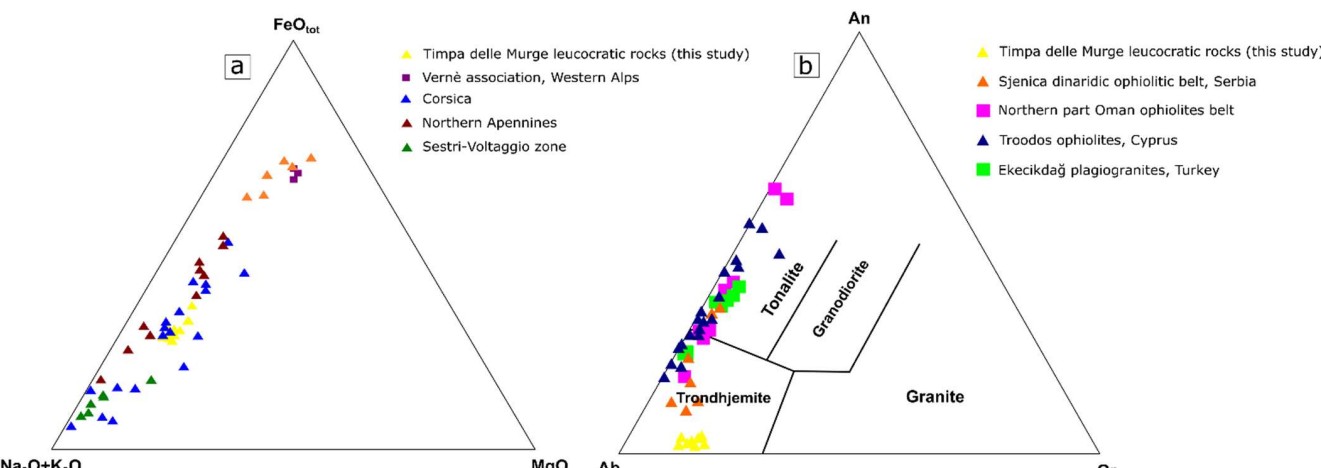

**Figure 11.** (**a**) AFM ternary diagram with compositional data for both leucocratic rocks of Timpa delle Murge and several Tethyan plagiogranites [15,17]; (**b**) Ab-An-Or ternary diagram for Timpa delle Murge leucocratic rocks and other Tethyan plagiogranites [1,18–21].

A further comparison was made by using the normative triangular An-Ab-Or diagram (Figure 11b) for Timpa delle Murge leucocratic rocks from the Tethyan area which are located in Serbia, Turkey, and Cyprus [1,18–21]. In general, the values of An in all the considered rocks from the Tethyan realm are very variable and range from ≈10 to ≈60%, with lower values observed in the Timpa delle Murge leucocratic rocks and Sjenica ophiolitic belt plagiogranites (Serbia) whereas higher values are relative to some samples of the Oman Northern ophiolites belt and Troodos ophiolites (Cyprus).

Figure 12 shows a Y-Nb binary diagram with data for the studied samples and plagiogranites from other areas [13,14,18,51,57,62]. The compositions of the studied rocks fall within the Volcanic arc and Syn-Collisional Granites field. In particular, the Nb concentrations of the studied samples from leucocratic rocks of Timpa delle Murge, are higher than those of most other Tethyan plagiogranites, in fact, they are comparable only with some samples of the Corsica, showing higher Nb content, and part of Tuscanian plagiogranites complexes, which, however, fall into the Ocean Ridge Granites field. In contrast, Y contents are consistently lower than those of all other considered plagiogranites.

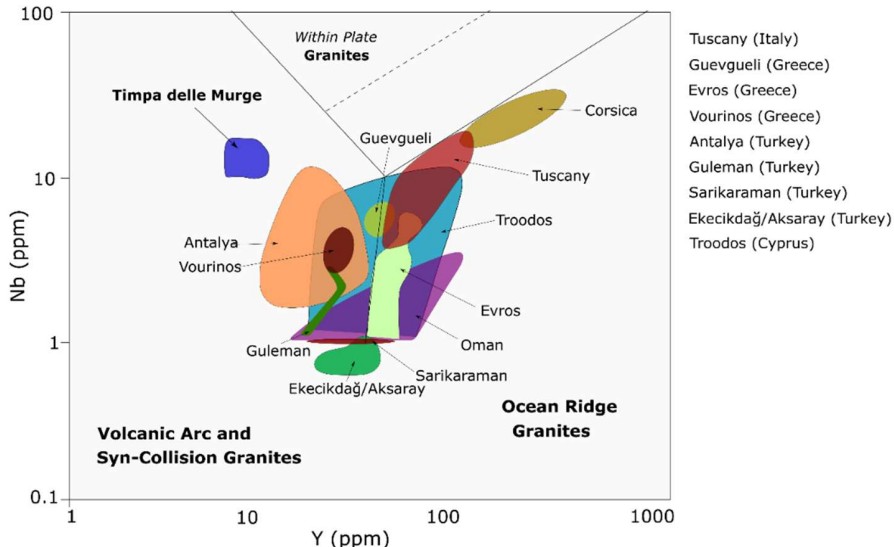

**Figure 12.** Y vs. Nb diagram of Timpa delle Murge leucocratic rocks and Tethyan plagiogranites displaying the fields of within plate granites (upper part), ocean ridge granites (on the right), and volcanic arc and syn-collisional granites (on the left) [13,14,18,51,57,62,63].

### 4.4. Tectonic Setting

Pearce et al. [57] classified granites based on their tectonic environment into volcanic arc granites (VAG), syn-collisional granites (syn-COLG), within plate granites (WPG), and oceanic ridge granites (ORG). This classification is based on discriminating elements such as Rb, Y, and Nb. In the Nb vs. $SiO_2$ diagram the Timpa delle Murge leucocratic rock fall into the VAG+COLG+ORG field (Figure 13).

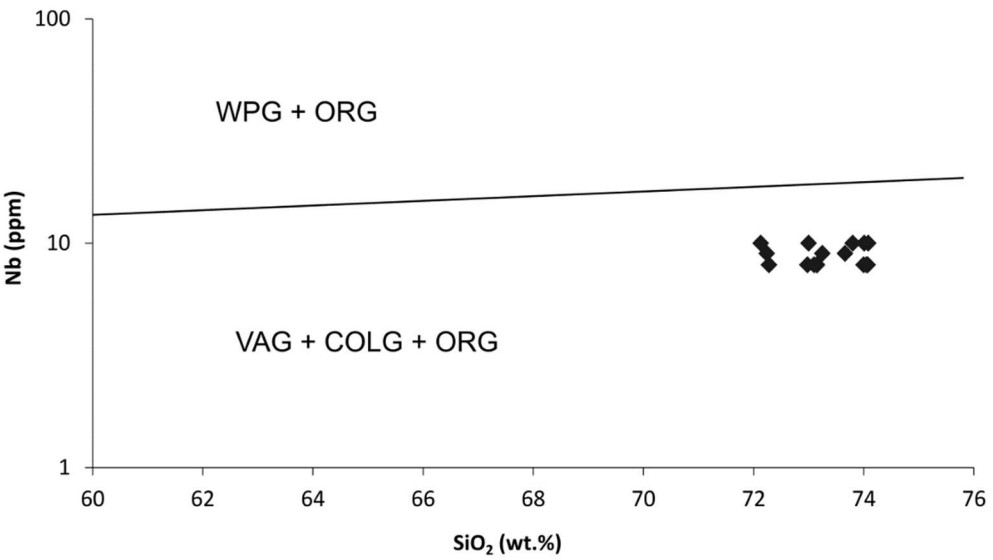

**Figure 13.** Nb (ppm) vs. $SiO_2$ (wt.%) diagram. The line divides the diagram into two fields which represent possible environments of granitoid formation. WPG: within plate granites; ORG: oceanic ridge granites; VAG: volcanic arc granites; COLG: collisional granites (modified after [57]).

Tectonic environment, for trondhjemites, is that of volcanic arc granites (VAG), as reported in Figure 9; in fact, Ta and Nb anomalies, relative to Th and Ce, are diagnostic of volcanic arc environments [54,63,64]. Furthermore, enrichments in K, Rb, Ba, and Th relative to Nb and Zr and depletions in Y are typical of volcanic arc granites [53].

The significant enrichment in Th is indicative of subduction zones [65,66]. The analyzed samples show Th/Y ratios between 0.89 and 1.7, much higher than that of MORB which is 0.03 [14]; this could indicate enrichment of magma with fluids or melts separated from the subducting plate [66,67].

Finally, enrichment in LILE and depletion in HFSE is interpreted as a consequence of subduction [68,69]. Magma composition is related to the interaction between the subducting plate and the mantle. The low Ti and Zr rates of the samples suggests low depths partial melting which, in turn, indicates low crustal thickness. Interaction of subducting plate and mantle induces change in magma composition and low concentration of elements coming from the mantle indicates a depleted mantle source [14]. In the analyzed leucocratic rocks, the lower concentration of mantle-derived elements, such as Nb, Ti, Y, and Yb, relative to the normalized compositions, evidence, in fact, a depletion of mantle in these components [14].

The origin of plagiogranites is mainly associated with two very different processes which are the fractional crystallization from a gabbroic source [70], and/or the partial melting of metasomatized gabbros [8,10]. The petrographic and the geochemical features of the Timpa delle Murge leucocratic rocks, suggest a model involving a rifting stage in a subduction setting, where the ophiolitic gabbros intruded the continental crust (Figure 14). This scenario promotes the partial melting and generation of crustal derived melts producing rocks of trondhjemite composition [71].

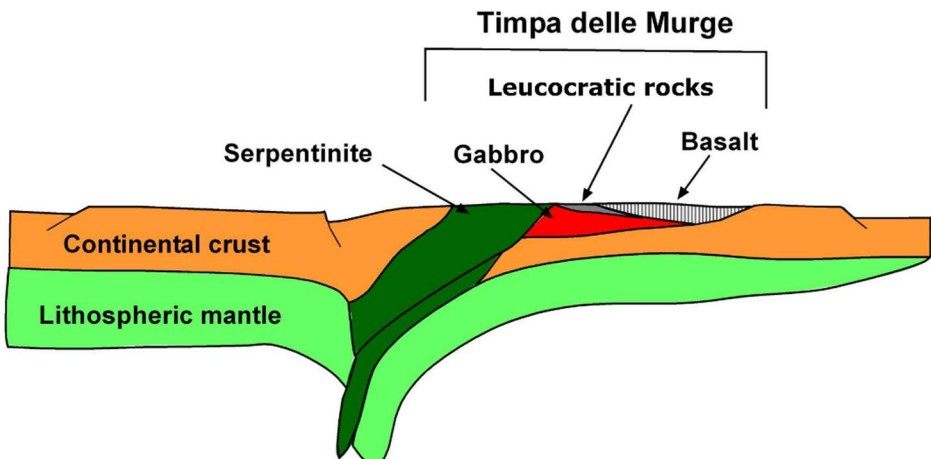

**Figure 14.** Schematic illustration showing the tectonic setting regarding the formation of the leuco-cratic rocks in the Pollino Massif.

## 5. Conclusions

The Timpa delle Murge ophiolitic sequence in the Northern Calabrian Unit is part of the Liguride Complex in the Pollino Massif and, for the first time, a petrographic and geochemical study were carried out on leucocratic rocks occurring in these ophiolitic sequences. These ophiolites do not show any subduction related high-pressure meta-morphism whereas the presence of gabbros and leucocratic suggests that the Northern Calabrian Unit experienced an intrusive activity that can be classified as subduction related.

The petrographic features of the analyzed samples showed a consertal hypidiomorphic granular structure. The observed mineralogical assemblage consists of plagioclase, quartz, feldspar, muscovite, biotite, and secondary chlorite. Their color index (M) is less than 10, in fact, the mafic minerals present are biotite and chlorite. These leucocratic rocks are classified as trondhjemites, in accordance with the IUGS classification.

Leucocratic rocks of Timpa delle Murge show high amounts of silica ($SiO_2$) with moderate alumina ($Al_2O_3$) and iron-magnesium (FeO-MgO) contents. Overall, the geochemical constrains, including both major oxides and trace elements are consistent with those from Western Alps, Northern Apennines, Turkey, Cyprus, Serbia, and Oman. The analyzed samples have higher potassium contents than the plagiogranites related to MORBs, this can be due to a partial plagioclase alteration to sericite. The Ab-An-Or ternary diagram shows that the analyzed leucocratic rocks fall within the trondhjemitic field and have subalkaline features, consistent with a possible differentiation from a tholeiitic gabbroic source mixed with crustal melts.

Spider diagrams exhibit compositional features which are consistent with fractional crystallization from a mafic magma as observed by several authors (Borsi et al. [16], Köksal et al. [18], and Milovanovic et al. [19]). The role of plagioclase fractionation is clearly evidenced by Eu depletion in an evolution from plagioclase-rich crystal mushes. Compositional data qualify these rocks as trondhjemites formed in a volcanic arc environment during continental extension as already observed for the Pollino Massif ophiolitic gabbros.

**Author Contributions:** Conceptualization, G.R.; data curation, M.P.; formal analysis, R.S.; methodol-ogy, S.L., L.B. and E.C.; supervision, G.M.; writing—original draft, G.R.; writing—review and editing, R.B. and G.M. All authors have read and agreed to the published version of the manuscript.

**Funding:** This research was supported by G. Rizzo and G. Mongelli grants (RIL 2019) and by Ente Parco Nazionale del Pollino funds.

**Data Availability Statement:** Data sharing is not applicable to this article.

**Acknowledgments:** This research was supported by funds related to an agreement between the Ente Parco Nazionale del Pollino and the Department of Science of the University of Basilicata. The authors would like to thank the reviewers and the academic editors for their helpful comments and suggestions which improved the manuscript.

**Conflicts of Interest:** For this research, all the authors declare no conflict of interest.

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
