# Peer review of "Petrography and Geochemistry of the Leucocratic Rocks in the Ophiolites from the Pollino Massif (Southern Italy)"

_minerals, doi:10.3390/min11111264_

Round 1
Reviewer 1 Report
The article is devoted to the study of petrographic and geochemical features of the plagiogranites of ophiolite belts in southern Italy in comparison with other regions in order to discuss their possible genesis and geodynamic conditions of formation.
The introduction clearly defines the tasks of the petrographic and geochemical study of plagiogranites of the ophiolite complex of rocks of the Pollino massif in Northern Calabria and comparison them with plagiogranites of other regions in order to define them in the geodynamic model of the Mediterranean belt.
- This section reflects the main geological structures of the studied area of the Southern Apennines. A fragment of the geological scheme of the Timpa delle Murghe region is also given, which shows the rocks of the ophiolite complex formed in the late Cretaceous-Oligocene. The position of the Pollino plagiogranite massif is not clearly shown. It is shown that intrusive formations, including plagiogranites, constitute an insignificant part of the ophiolite complex.
- The factual materials are 16 plagiogranite samples taken in the gabbroids and basalts contact zone. At the same time, the form of occurrence of plagiogranites and their contact relationships are not clear. The main used instruments, equipment and methods of petrographic and geochemical research sufficient to solve the problem are presented.
- Results and discussion
The authors’ petrographic studies show that according to the mineral composition and the constructed normative diagrams Ab, An, Or demonstrate that plagiogranites have undergone metamorphic transformations of the greenschist facies and correspond to trondchemites. In the reviewer’s opinion this conclusion sounds reasonable.
In table 1, the chemical composition of plagiogranites represents the iron in the form of Fe2O3, and the value of FeO is not given, while i Fig. 7 shows the FeO / MgO ratio (wt%). We still need to adjust Fe2O (?).
The distribution of geochemical elements in plagiogranites is given in Table. 1. The diagram (Fig. 6) shows that plagiogranites belong to the subalkaline series based on the high content of Na2O, which is typical for this group of rocks. However, in general, in terms of the content of Na2O + K2O (about 6 wt.%), plagiogranites are characterized by a general low alkalinity and in other parameters (Fig. 7) they correspond to the tholeiite series. The general low alkalinity of plagiogranites is typical for other regions (Bolshoi Altai, 1998; Kuibida et al., 2019). The general conclusion of the authors about designating the Pollino massif plagiogranites as oceanic plagiogranites is sufficiently substantiated by factual material in the reviewer’s opinion.
The manuscript provides a comparative characteristic of plagiogranites with similar rocks from other regions of the Mediterranean and notes similarities and differences. It would be desirable to show whether there is any mineralization in the studied plagiogranites. For example, it can be noted that plagiogranites of the West Kalbinskaya zone of East Kazakhstan of the collisional type (C3) are characterized by an increased content of Sr, Li and are accompanied by gold mineralization.
In general, the manuscript was made at an utterly high scientific level, contains fundamentally new information on the petrographic and geochemical characteristics of oceanic plagiogranites, and makes a certain scientific contribution to the understanding of the petrology of granitoid magmatism. After minor corrections, it is recommended for publication.
Author Response
Dear Reviewer 1,
Thanks a lot for the suggestions that will help to improve the ms.

Reviewer 2 Report
The paper deals with a first petrographic characterization of the plagiogranites of the Pollino Massif ophiolites, in southern Italy. It is a work with a “classical” and solid approach, based on petrography and WR geochemistry, which fully falls within the topics covered by Minerals. The paper is well structured and provides significant data for the petrological interpretation of these rocks, compared with similar occurrences in different locations of the Thetyan realm. In the text, written in general in good English, I found only some minor ambiguities and inconsistencies: figures, tables and diagrams are clear and, in my opinion require only minimal changes. See the attached file for my observations and some suggestions.

Author Response
Dear Reviewer 2,
thank you so much for the suggestions that will help to improve the ms

Reviewer 3 Report
The manuscript deals with plagiogranites from an ophiolite sequence in southern Italy. The topic is clearly of interest to many readers. The conclusions that these rocks originate from fractional crystallization of mafic magma is reasonable. This is also true for the geotectonic assignment of the plagiogranites to a collisional setting.
The presentation is a bit wordy and repetitive and mainly descriptive. The main conclusions are not even mentioned in the abstract. The English definitely needs improvement (I have made a number of suggestions in this respect). Nevertheless, the manuscript can be published after moderate revision.

Author Response
Dear Reviewer 3,
thanks a lot for the suggestions that will help us to improve the ms.
